Review
# Recent approaches in computational modelling for controlling pathogen threats

John A Lees[1], Timothy W Russell[2], Liam P Shaw[3,4] , Joel Hellewell[1]

In this review, we assess the status of computational modelling of pathogens. We focus on three disparate but interlinked research areas that produce models with very different spatial and temporal scope. First, we examine antimicrobial resistance (AMR). Many mechanisms of AMR are not well understood. As a result, it is hard to measure the current incidence of AMR, predict the future incidence, and design strategies to preserve existing antibiotic effectiveness. Next, we look at how to choose the finite number of bacterial strains that can be included in a vaccine. To do this, we need to understand what happens to vaccine and non-vaccine strains after vaccination programmes. Finally, we look at within-host modelling of antibody dynamics. The SARS-CoV-2 pandemic produced huge amounts of antibody data, prompting improvements in this area of modelling. We finish by discussing the challenges that persist in understanding these complex biological systems.

## Introduction

Computational infectious disease modelling is the attempt to approximate the real-world biological processes of pathogen transmission, control, and evolution using mathematical and/or simulation-based techniques. In this review, we provide an overview of three distinct branches of disease modelling and consider the methods, approaches, and challenges within them. First, we explore the difficulties of modelling the problem of antimicrobial resistance (AMR). Correctly understanding the biological mechanisms driving AMR is highly complex and involves many pathogens and demographics, which makes accurately predicting changes in the prevalence of AMR difficult to achieve using models. Nevertheless, the outputs of these AMR prevalence models are fed into further models that try to predict the long-term health and economic impacts of AMR at a global scale.

We next consider the challenges of estimating how vaccines will alter subsequent pathogen evolution, focusing on the bacterial colonisers of the upper respiratory tract, *Neisseria meningitidis* and *Streptococcus pneumoniae*. The modelling in this section predicts the dynamics of competing bacterial strains that result from vaccinating human populations. Finally, we consider models of within-host immune response, primarily for COVID-19. Traditionally, one difficulty in modelling viral or antibody kinetics was a lack of high-quality data. However, for COVID-19, and an increasing number of other pathogens, modellers have access to more and better data than ever before. New approaches to modelling antibody kinetics must be developed for modelling data of a quality and quantity that was previously thought unobtainable.

These three examples were chosen to give a sufficient variety in the spatial and temporal scope of the modelling work. We cover changes in the antibodies levels within one person over a matter of months, changes in the bacterial strains within a vaccinated population in the years following a vaccination campaign, right through to global estimates of the impact of worsening AMR over the next few decades. Despite their differing scopes and methodologies, we will try to identify broad trends and challenges shared across these three fields of modelling. Where possible, we will also link our discussion in each of these examples to the COVID-19 pandemic, where computational modelling was used extensively and brought infectious disease modelling to the forefront of scientific and public consciousness.

The original emphasis of modelling was on using mathematical analysis tools to understand the qualitative behaviour of a "model," defined by a system of equations. Knowing how the behaviour of a model changes over time for certain combinations of parameter values can lead to useful qualitative insights regarding the real-world biological system, insights that might not be obvious without the use of models as an explanatory tool. In contrast, we think that contemporary modelling places greater emphasis on statistical and computational machinery, allowing the available data to guide the form of the equations within the model. The aim is to use the ability of the model to predict the data as evidence to accept or reject different model structures, each ideally corresponding to different hypotheses about the biological system being studied.

[1]European Molecular Biology Laboratory, European Bioinformatics Institute, Wellcome Genome Campus, Hinxton, UK [2]Centre for Mathematical Modelling of Infectious Diseases, London School of Hygiene & Tropical Medicine, London, UK [3]Department of Biology, University of Oxford, Oxford, UK [4]Department of Biosciences, University of Durham, Durham, UK

Correspondence: joel@ebi.ac.uk

To give an example of the changes in the approach to infectious disease modelling over time, we will briefly turn to malaria modelling. An early and influential malaria transmission model was the Ross-MacDonald model that was in development from the 1950s to the 1970s. The Ross-MacDonald model began with modellers trying to inscribe their existing theories of the process of malaria transmission in the structure of mathematical equations. From a few basic assumptions of how transmission works, an equation was derived for the reproduction number (R0, the average number of secondary infections for each new infection). Despite the fact that this model was not fitted to data on malaria cases, it provided considerable insights into strategies that might control malaria transmission.

In the Ross-MacDonald model, R0 turns out to have a linear relationship to all model parameters, except for the biting rate on humans which has a quadratic effect, and mosquito survival which has an approximately cubic effect (Smith et al, 2012). Therefore, reducing the biting rate and mosquito survival has a larger impact on reducing R0 than the other parameters. This model-derived insight supported early malaria eradication efforts through indoor spraying with the insecticide DDT. It is also the rationale behind current malaria prevention tools: insecticide treated bed nets that both kill mosquitoes and provide a physical barrier preventing them from biting people. Although this result has had a positive effect on malaria control to date, the concern that motivated a move towards a more data-led contemporary modelling approach over time is that this result, derived from the mathematical structure of the model, is only applicable if the model is a "good enough" approximation of real-world malaria transmission. A contemporary example of malaria transmission modelling considered multiple structural forms for the model equations, choosing between them based on their ability to best explain the data from experimental settings and long-term malaria incidence trends over multiple countries (White et al, 2011).

Returning to speaking of modelling more generally, contemporary modelling practice usually involves some form of data-based model selection from a set of biologically plausible candidate models. The model fit is evaluated predominantly on the model's ability to predict the values of the same data that it is fitted to, combined with a penalty for model complexity that aims to prevent "overfitting" the model to finite data. This sort of model selection only measures predictive power compared to the other candidate models, the best model of the set may still predict poorly. Two different modelling approaches emerge here: the "scientific" approach, which is more concerned with linking model development and model fitting to answering scientific questions, and the "pragmatic" approach, which searches for the model with the best objective predictive accuracy for use in practical decision making (Navarro, 2019). The two approaches may not always select the same "best" model. Which approach is followed should depend on what the model is being developed for and what sort of questions it is being designed to answer.

Most of infectious disease models split the population into separate compartments that represent the different states of disease (susceptible, infected, recovered, etc.). The flow of people between the compartments can be deterministic, typically represented by a system of coupled differential (or difference) equations; or stochastic, represented as a set of rules describing the probability that individuals move between compartments over time. Model complexity can vary depending on the detail of available data and the structure required to model a given problem. Often models further split the population by age, spatial structure, or behaviour.

Alternatively, individual-based models simulate individuals following rules describing the probability that they transition between disease states. Here, each individual's specific disease state is tracked rather than the total number of individuals within each disease state like in compartmental models. This approach can provide more granular estimates than compartmental modelling, especially if individual-level data are available to parameterise the model accurately. Because of the great interest in SARS-CoV-2, and the resulting colossal data collection, a number of recent studies modelled the spread of SARS-CoV-2 using detailed individual-level and/or household-level data. For example, Ferretti et al (2023), arrived at time-dependent estimates of the probability an individual would be infected with SARS-CoV-2 after they were exposed—a key quantity crucial for parameterising other models of COVID-19 transmission. We now turn to our case studies, before discussing the common threads alluded to across the the studies.

**Modelling of AMR**

AMR refers to the general problem of microbial pathogens which are able to withstand treatment with antimicrobials. By convention, AMR is often used to refer to the problems of resistance to antibiotics in bacteria, but it should be noted that the term can also be used to encompass resistance in fungi (antifungal resistance) and viruses (antiviral resistance). AMR modelling is more disparate than for other pathogen threats (e.g., viral epidemics) because it is a diffuse cross-pathogen threat. For our purposes, we can treat modelling of AMR as falling into three broad areas: *calculating* the levels of AMR, *explaining* why we observe those levels, and *informing* us how to reduce them. Rather than aiming to be comprehensive, we first introduce the general features of AMR before giving some clear case studies for each area.

The full complexity of AMR is beyond the scope of this review, but it is a diverse set of phenomena driven by diverse biological mechanisms (Darby et al, 2023). Depending on the question, modelling of AMR pathogen threats may not need to engage with the genetically determined complexity of AMR. For example, in November 2016, an outbreak of typhoid fever in the Sindh region of Pakistan of an extensively drug-resistant (XDR) form of *Salmonella* Typhi caused global concern (Klemm et al, 2018). XDR Typhi quickly became the dominant cause of typhoid fever in Pakistan: from no cases in 2017 to 50% of cases in 2019 (Nizamuddin et al, 2021). The presence of XDR Typhi could be determined from its phenotypic resistance profile and so it could be modelled as a new and separate pathogen to "regular" Typhi. The underlying biological causes of the resistance—a combination of new resistance genes (including on a plasmid) as well as mutations in a chromosomal gene—did not need to be incorporated into models of its spread. A modelling analysis assessed the global risk of further outbreaks of XDR Typhi using air travel data in combination with reported cases, finding that countries with more passengers arriving from Pakistan

were far more likely to have cases (Walker et al, 2023). This analysis highlighted the probable existence of unreported cases in countries with high air traffic with Pakistan (Saudi Arabia, Turkey, and Malaysia) as well as countries at high risk of XDR Typhi outbreaks. Afghanistan was judged at high risk of XDR Typhi outbreaks given its already high incidence of typhoid cases and high connectivity to Pakistan.

This analysis had clear public health implications, but a common reason for developing computational models of AMR is to better understand the drivers of resistance to help us work out how to reduce it. The fact that AMR is inherently an ecological problem makes this more challenging than modelling a single pathogen with an "SIR"-type model of transmission. To take an important example: *Escherichia coli* is a diverse species, with subtypes including common gut commensal strains, but also phenotypically quite different strains that cause opportunistic extraintestinal infections. Both subtypes may be either sensitive or resistant to a given antibiotic. Resistance genes can be carried and exchanged between both commensal and pathogenic strains—or even with other bacterial species—and resistance to one antibiotic may be correlated with resistance to another. The boundaries of what needs to be included in a model to accurately capture the system are unclear. Not only that, but the underlying data quality is often poor because of a bias towards sequencing resistant isolates, varying regional surveillance capacity and sometimes a lack of standardisation between laboratories. This is part of the reason that AMR modelling is less advanced than for other pathogen threats. As we argue in what follows, modelling must tackle these data challenges now rather than waiting for better data.

### Calculating the incidence of AMR

How much of a problem is AMR? Answering the question requires modelling. Well-known statistics about AMR are often the products of models. For example, the much-cited O'Neill report commissioned by the British government stated that AMR could cause 10 million deaths a year by 2050 (O'Neill, 2016). This alarming statistic was based on analysis commissioned from two consultancy firms, KPMG and Rand, which do not go into much methodological detail (KPMG LLP, 2014). Criticising the "10 million deaths by 2050" figure, de Kraker et al noted it came from a hypothetical scenario where infection rates doubled and resistance rates rose by 40 percentage points then remained stable—strong assumptions without data supporting them (de Kraker et al, 2016). De Kraker et al argued that "modeling future scenarios using unreliable contemporary estimates is of questionable utility."

More recently, a Global Burden of Disease study tried to estimate the current burden of AMR—or, strictly speaking, the burden of 23 key pathogens and 88 pathogen-drug combinations across 204 countries in 2019 (Ranjbar & Alam, 2023). By considering the counterfactual scenario where every resistant infection was instead a sensitive infection, Murray et al (2022) aimed to estimate the number of deaths that were directly attributable to AMR. After a complex modelling process involving sub-models for each pathogen, their final estimate was 1.27 million deaths, with a 95% uncertainty interval of 0·91 – 1·71 million derived by propagating uncertainty through models and taking quantile ranges from the posterior distribution of parameters.

It is worth highlighting how much modelling is behind this headline figure. First, because causes of death are rarely coded using pathogen or resistance profile but rather by infectious syndromes with diverse underlying microbial causes, the authors used models to relate syndromes to pathogens. Second, poor data availability meant that the authors used models to generate data for the next stage of modelling. Their final estimates are necessarily built up from a succession of models, with "10 estimation steps that occur within five broad modelling components" that hierarchically create inputs for the next models: from models at the level of infectious syndromes, to case-fatality ratios, pathogen distributions, the fraction of resistance, and finally the relative risk of resistant versus susceptible infections. Making such a complex set of models across pathogens is clearly a difficult task and models may miss aspects known to be important for a particular pathogen. For example, the model for *S. pneumoniae* did not account for serotype replacement after vaccination.

Murray et al (2022) acknowledged significant limitations, including a lack of data from many low- and middle-income countries. Indeed, 19 countries had no available data at all for any aspect of the study's modelling. This lack of data is particularly problematic given that, where data are available, it suggests that AMR is much more of a problem in low- and middle-income countries. Data scarcity—because of systematic global inequalities—has been highlighted again and again in the context of AMR. However, even where we have good data, the situation is far from clear because our ability to explain resistance with simple models is poor.

### Explaining observed levels of resistance

In its fundamentals, AMR is an evolutionary process: an effective antimicrobial exerts a selective pressure for resistance. Put so starkly, AMR might appear like a trivial problem to model. We know that increased usage of an antibiotic should lead to more prevalent resistance. But despite this, predicting population-levels of resistance is surprisingly difficult. To take a simple example, consider a pathogen with two subpopulations: a sensitive strain and a resistant strain. Assuming the resistant strain is fitter in the presence of antibiotics, this simple model would predict competitive exclusion: there will be a level of antibiotic prescribing below which the sensitive strain dominates and above which the resistant strain dominates. But empirically, we usually observe the persistent coexistence of sensitive and resistant strains over many years, such as for *Streptococcus pneumoniae* (Lehtinen et al, 2017; Blanquart, 2019).

Many possible model structures can reproduce some form of this coexistence. One early effort used a Monte Carlo simulation of 10,000 human hosts that could exchange bacteria with the environment, in effect producing a "migration-selection balance" where an influx of sensitive strains balanced the selection of resistant strains (Levin et al, 1997). Although many other model structures can also reproduce coexistence patterns, one group of authors argued that models should have no intrinsic mechanism that promotes stable coexistence of strains that are otherwise indistinguishable. Otherwise, models can artificially increase the conditions under which coexistence occurs, rather than explaining it realistically (Lipsitch et al, 2009).

The same group of authors compared related models, where a host could be infected by both sensitive and resistant strains at the same time (represented by two equally sized compartments) concluding that within-host interactions play a more important role in coexistence than treatment and contact heterogeneity (Colijn et al, 2010). A more recent study criticised this model, arguing that the subcompartment assumptions (amounting to equal abundance within a host) inhibited coexistence by reducing the scope for within-host competition (Davies et al, 2019b). Those authors argued for a "mixed-carriage" model that explicitly tracks within-host strain frequencies, arguing that this outperformed the previous model when capturing the relationship between national antibiotic consumption and resistance prevalence in *E. coli* and *S. pneumoniae*. However, Davies et al did not use any real within-host data. Similar models accounting for maintained structure and separation in the host population, or structure within the pathogen population, have also been adapted to attempt to explain observed frequencies of multiple resistance (Lehtinen et al, 2019).

Some modelling efforts do not try to explain in term of mechanisms but combine antibiotic prescribing data (from electronic health records) with resistance data from longitudinal surveys and look for time–series correlations. These correlations can be modelled with elastic net regularisation and generalised boosted regression models. Such models have highlighted that use of one antibiotic can correlate with resistance to other antibiotics, either because of shared resistance mechanisms or the genetic linkage of resistance genes. One analysis of antibiotic use in primary care in England found that regional levels of resistance to trimethoprim (a sulfonamide antibiotic) were better explained by prescribing levels of amoxicillin than by prescriptions of trimethoprim itself (Pouwels et al, 2018). Amoxicillin prescribing was also correlated with resistance to ciprofloxacin, a different class of antibiotic (Pouwels et al, 2019).

AMR varies seasonally, and this can be captured using oscillatory models with a period of 1 yr. A recent study of AMR in the USA showed that resistance to all antibiotic classes was most correlated with the use of penicillins and macrolides, the most highly prescribed antibiotic classes (Sun et al, 2022a). Usage typically peaks in winter, suggesting that seasonal selection is dominated by only some antibiotic classes. There are many further complications of the use-resistance relationship: for example, high levels of resistance can lead to reduced use of an antibiotic because it is less likely to be effective. Reducing antibiotic prescribing is one obvious action to take to reduce AMR, but understanding how effective our actions will be requires modelling.

### Informing appropriate action

As well as helping us to understand the underlying evolutionary processes that produce patterns of AMR, computational models would ideally help us to use antibiotics more effectively by forecasting what might happen with different scenarios of use.

To take just one example, it might seem reasonable that a new antibiotic should be held in reserve to prolong its clinically useful lifespan. A recent study used a model to inform how a hypothetical new antibiotic for resistant infections would best be deployed to increase the time taken for resistance to reach a 5% prevalence threshold (Reichert et al, 2023). The researchers focused on

*Neisseria gonorrhoeae* infections in men who have sex with men. They used a compartmental transmission model where men are stratified into three levels of sexual activity and can all move between different infection states, both symptomatic and asymptomatic (Fig 1A).

Reichert et al then investigated the case where default treatment with antibiotic A could be altered thanks to the availability of a new antibiotic B. They compared four different strategies: immediate random allocation (A or B), combination therapy (A and B), a gradual switch to random allocation, or a reserve strategy where B was held in reserve and then used in place of A after resistance to A had reached a 5% prevalence threshold. Modelling showed, counter-intuitively, that this latter "reserve" strategy speeded up resistance. The best strategy was to deploy B immediately in combination therapy (Fig 1B). Others have also argued that combination therapy is the best approach to reduce AMR, including an elegant in vitro experimental model using well plates to simulate individuals in a hospital population (Angst et al, 2021). AMR may be driven more by the wide distribution of use than its intensity (Olesen et al, 2018), and some have argued that for *Neisseria gonorroeae* issuing prescribing guidelines based on local prevalence thresholds could reduce cases and prolong the lifespan of antibiotics (Yaesoubi et al, 2022).

Other control measures such as vaccination can affect AMR. Modelling of the "bystander effect," where microbes are exposed to antibiotics despite not being the target organism, suggests that vaccination programmes for *S. pneumoniae* also result in a similar reduction in total antibiotic exposure for *Staphylococcus aureus* and *E. coli* as for *S. pneumoniae* (Tedijanto et al, 2018). Vaccination can also affect competition between strains in a population. One modelling effort showed that four different models with different assumptions about the mechanisms of AMR evolution explained penicillin resistance in *S. pneumoniae* across Europe equally well, leaving it unclear whether a hypothetical vaccine program would increase, decrease, or have no effect on AMR levels (Davies et al, 2021).

AMR is conceptually simple. In practice, the examples in this section show that making realistic models takes care. In 2019, one group of scientists wondered "do we know enough?" to effectively use modelling to inform policy—concluding that, for the most part, the answer was no (Knight et al, 2019). One indication of these challenges is that, in contrast to other pathogen threats, there have been barely any attempts at real-time forecasting applied to AMR, that is, where current data are used to project forward, rather than a retrospective analysis (Pei et al, 2023).

In summary, our understanding of AMR as a pathogen threat remains broad-brush. We have a qualitative understanding that antibiotic use selects for resistance, but our quantitative understanding of that relationship is poor. Furthermore, although antibiotics are a fundamental part of modern medicine they are used very differently in different healthcare settings—for example, in some countries antibiotics require a prescription, but in others, they are available without one. A pernicious aspect of data availability is that modelling is heavily biased towards countries where there is a lot of data; in these countries, AMR is typically less of a threat. Models can lead to recommendations for the better use of antibiotics, but it is worth stressing that in many global settings

## A  Compartmental model

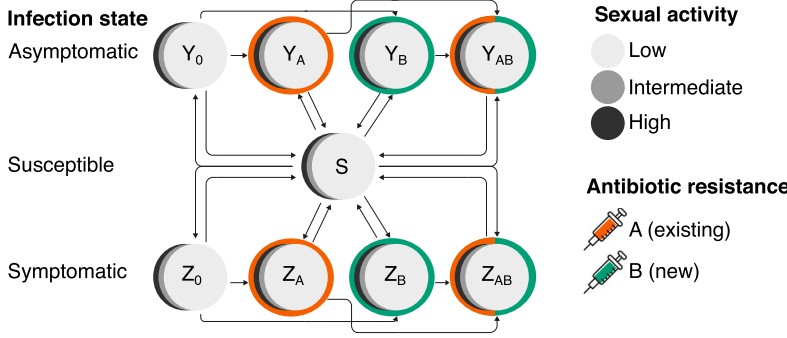

## B  Comparison of treatment strategies

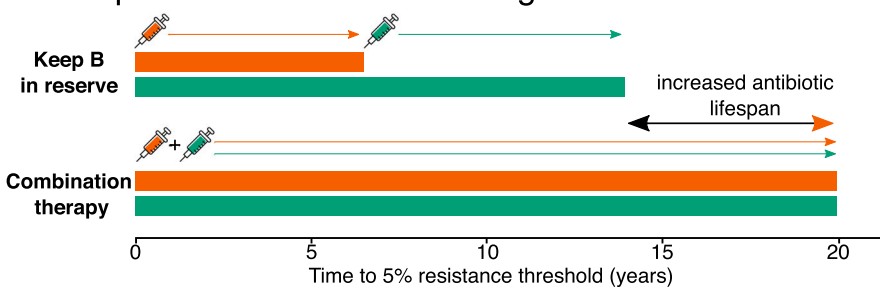

**Figure 1.  A model of N. gonorrhoeae infection in men who have sex with men can be used to compare different strategies for the introduction of a new antibiotic.**
**(A)** Schematic of the transmission model, adapted from Reichert et al (2023). Men can be either susceptible (S), asymptomatically infected (Y), or symptomatically infected (Z). Infected men can carry a strain that is resistant to either none, one, or both antibiotics A and B (subscripts). Each compartment contains three subcompartments for men with different levels of sexual activity. The flow between compartments is illustrated by arrows. **(B)** Illustrative comparison of predictions arising from the model for two possible treatment strategies: keeping B in reserve until resistance levels to A reach a 5% prevalence threshold then switching to B or using A and B in combination from the start. The model predicts a substantial delay in the onset of resistance with the combination strategy, resulting in increased lifespan for both antibiotics. Data taken from Table 2 of original paper. For full details, see the original paper.

the single most important intervention to reduce the threat of AMR is improved water sanitation and hygiene to reduce infectious disease in general.

### Modelling of population dynamics to design vaccines
The dynamics of infection and vaccination are non-linear and difficult to predict even with large amounts of data (Nokes & Anderson, 1988; Bosse et al, 2022). The use of mathematical models to predict the health and economic impacts of vaccination campaigns has a long history, stretching back to the introduction of the very first vaccine against cowpox in the eighteenth century: the impact of vaccination was predicted based on the life expectancy of the population (Bernoulli, 1766). By the 1980s, mathematical models of infectious disease transmission with complex and more realistic descriptions of infectious disease dynamics had been developed for many pathogens (Anderson & May, 1985; Anderson et al, 1992), offering qualitative insights into vaccination strategy, that is, the "scientific" approach: what fraction of the population needs to be immunised; what is the effect of waning immunity; what are the effects of vaccine escape strains? (Scherer & McLean, 2002).

The advent of cheap and widely available computation starting in the 1990s precipitated a shift from analytically tractable infectious disease models to more complex differential equation models and individual-based models which have to be solved numerically (i.e., by computer simulation) (Ferguson et al, 2003). These more complex models could promise overall better predictive accuracy of dynamics including error estimates on model outputs—the "pragmatic" approach. These advances led to a variety of helpful outputs including more quantitative assessment of disease burden after vaccination, estimating the cost efficacy of vaccination, and

testing the predicted success of different vaccination dose strategies (Sonabend et al, 2021; Ryman et al, 2022). Some of the most recent efforts have been able to combine complex models of 10 vaccines used in 100 countries to estimate their total impact (Li et al, 2021; Toor et al, 2021). These uses of models to inform vaccination require the testing of multiple potential models against biological knowledge and data, validation of model fit over historical data, the use of forecasting to predict future dynamics, and the ability to effectively and accurately incorporate vaccination policy and effect into the model.

These computational advances in modelling coincided with the development of new vaccination technologies such as conjugation (Eskola et al, 1990; Kelly et al, 2004; Trotter et al, 2008), followed by the subsequent licensing of many more vaccines for use in humans. Combined with accurate and flexible models which could be fit and tested at scale, this created a growing appetite for the use of modelling to design vaccine strategy (Christen & Conteh, 2021). Consequently, computational modelling approaches are now routinely used both before, during and after vaccination programmes to optimise their effectiveness, monitor their ongoing success, and determine whether modifications and improvements are possible.

This is only a brief overview of using computational modelling to design vaccine strategies as this topic has been reviewed in detail elsewhere (Nokes & Anderson, 1993; Scherer & McLean, 2002; Reid et al, 2019; Bershteyn et al, 2022; Wagner et al, 2022). For this review, we choose to focus on an emerging computational modelling technique expected to be used in the design of a number of upcoming vaccines: the development of population genomic models for multi-strain pathogens.

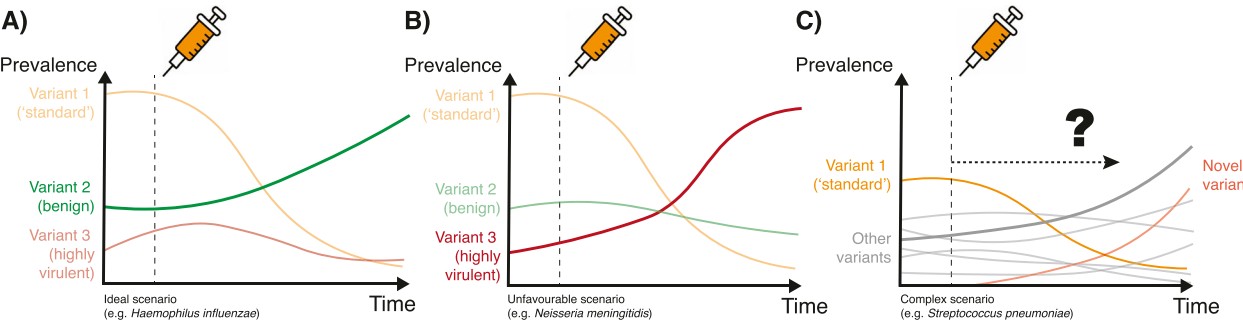

**Figure 2. An illustration of vaccine-induced population dynamics.**
In each case, an effective vaccine is rolled out against the dominant strain, in yellow. After vaccination, cases of the dominant strain decrease. However other strains, having less competition for hosts, are able to fill the "gap" which has been left in the population. **(A)** In the first example (panel (A)), a benign non-disease strain (green) fills this gap and therefore total cases of disease in the population are reduced (illustrative of *H. influenzae*–vaccination against serotype B). **(B)** In the second example (panel (B)), a more virulent strain is able to take over, ultimately meaning the vaccine has no effect, or even a detrimental effect on, total disease burden (illustrative of *N. meningitidis*–yellow strain serotype C; red strain serotype W). Modelling can be used to try and predict which situation will occur. **(C)** In the third example (panel (C)), representing a complex population with many strains (illustrative of Streptococcal pathogens), vaccination against all strains is not feasible. Modelling can be used to predict which non-vaccine variants will come to dominate, whether newly emerged strains will be successful, and optimise the initial vaccine formulation.

**Multivalent vaccination against pathogen subtypes** Pathogens are under strong evolutionary pressure to diversify the array of antigens they present to the host. As previously infected hosts develop immunity against antigens they have encountered before, a biochemically different antigen which partially or totally evades this immunity provides a significant transmission advantage in an exposed population. Most pathogens have variable surface antigens, and consequently multiple strains. Well-known examples of subtypes of a pathogen include hemagglutinin (H) and neuraminidase (N) antigens of influenza viruses, and the Omicron variants of SARS-CoV-2[1]

It is practical to design vaccines which immunise against many subtypes at once, known as multivalent vaccines. The immediate questions are: which variants should be included in the vaccine; should these change over time or for different regions; in what proportion should each variant be present in the dose; how should protection against targeted variants be evaluated? Given the non-linear nature of transmission dynamics and the increased complexity of these systems, computational modelling of transmission is once again a powerful tool to address vaccine design.

For some pathogens, these decisions are straightforward. Dengue virus vaccines must immunise against all four subtypes, or else, the risk of severe disease from an unimmunised subtype increases (Thomas, 2023; Kallás et al, 2024). *Haemophilus influenzae* vaccines need only immunise against serotype b (Hib), which is the one serotype which causes severe disease (Kelly et al, 2004) (Fig 2A). Bivalent COVID-19 vaccines contained a 50:50 ratio of the ancestral (Wuhan) spike protein sequence and the subsequently dominant Omicron (BA.1) variant spike protein, which evades much of the prior immunity from infection or immunisation from the ancestral protein (Chalkias et al, 2022). However, the rapid transmission of COVID-19 compared with vaccination development cycle and the presence of one dominant antigen in the population make reactive design of multi-valent COVID-19 vaccines challenging.

What type of pathogen is a good candidate for using computational modelling to optimise design of a multivalent vaccine? There would need to be many subtypes against which the vaccine could be made. The vaccine must be effective against each targeted subtype. To be economically practical, vaccination would have to be expensive so that optimisation benefits outweigh the costs of changing vaccine formulation, and rates of variant spread must be slower than vaccine roll-out.

One such candidate is the bacterium *N. meningitidis*, a commensal organism of the upper respiratory tract and occasional cause of bacterial meningitis. The species has 12 known serogroups, six of which cause invasive disease (Parikh et al, 2020). Complex dynamics of these serogroups have been observed. A major cause of disease was serogroup C, against which an effective conjugate vaccine was introduced in many high-income countries starting in 1999 (Fig 2B). A subsequent 20-fold reduction in serogroup C cases was observed over the next 5 yr (Trotter & Ramsay, 2007). However, starting in 2009, the incidence of a "hypervirulent" serogroup W increased, driven by a single genetic strain (Knol et al, 2017). Serogroup Y also increased in prevalence (Ladhani et al, 2012).

Ecologically, the decrease in serogroup C has left a "hole" in the infected host population which other strains may be able to fill. Whether the increase in prevalence of W and Y was caused in part by the vaccination against serogroup C is unknown, but the subsequent emergence and increase in a more virulent strain after an intervention is an example of a result which is hard to predict without a formal model. Fortunately for this pathogen, vaccines against all virulent serogroups are now in national immunisation programmes, leaving a "hole" that will hopefully be filled by benign strains or species (Ladhani et al, 2016).

**Modelling to improve the design on multi-valent pneumococcal vaccines** *Streptococcus pneumoniae* has over 100 serotypes, more than 1,000 strains and an effective but expensive vaccine (Croucher

---

[1]We use the terms subtype, strain, serotype, and variant interchangeably to refer to pathogens of the same species with different antigens, although for each species there tends to be one preferred term.

et al, 2018; Lo et al, 2019). This vaccine is effective at reducing carriage in children, protecting adults indirectly. The pneumococcal conjugate vaccine (PCV7) originally contained the seven serotypes which caused the most invasive disease in children in the USA. However, a decade after rollout, even with an addition of five further serotypes in the PCV13 vaccine, non-vaccine serotypes had increased in prevalence to a degree that the total burden of disease had returned to pre-vaccine levels in older age groups (Ladhani et al, 2018) (Fig 2C). A vaccine with 100 serotypes ("PCV100") will likely never become a reality (Løchen et al, 2020); the highest valency currently available is PCV20 (Essink et al, 2022).

Given the expectation of ongoing serotype replacement eventually restoring the total number of infections, how should the serotypes in the vaccine then be chosen? A good criterion would be long-term reduction in burden of disease, potentially when accounting for age. Minimising rates of AMR may also be desirable, so when disease does occur, it can be effectively treated. To do this, we need multi-strain computational models which can accurately forecast population dynamics after vaccination, test the effects of putative vaccine formulations, and predict at least strain resolution.

Models of multi-strain pathogens have existed since the 1990s (Gupta et al, 1996; Andreasen et al, 1997; Gog & Grenfell, 2002). Multi-strain models typically have more states and parameters than a single-strain model, presenting computational, biological and data-base challenges. Some of these early models noted the problem their predicted complex dynamics would pose for multivalent vaccines (Gupta et al, 1998).

The first modelling approach used for this task proceeded by making the simplifying assumptions that both disease rates from and proportions of the non-vaccine serotypes would remain unchanged after vaccination (Nurhonen & Auranen, 2014), the authors were able to derive an analytic expression for the vaccine formulation which minimises invasive disease burden at the current time.

A subsequent analysis used whole genome datasets from four populations collected spanning different vaccine introduction schedules (Corander et al, 2017). This modelled postulated frequency-dependent selection as a mechanism to maintain gene prevalence. By adding this genomic component to the model, superior predictive power of future strain prevalence after vaccination was achieved (Azarian et al, 2020). Subsequent extensions to this model searched over a space of possible vaccine formulations to find multi-valent combinations which would minimise burden of disease and AMR, specifically for a given population with historical carriage data (Colijn et al, 2020).

As well as these very specific quantitative predictions this model creates a more qualitative suggestion: adding serotypes to the vaccine has diminishing returns, and instead vaccinating adults with serotypes not in childhood vaccines would have more impact. This is also supported by the shared genetics of adult and childhood strains (Kremer et al, 2022). This is a simple design suggestion backed by data, modelling and also sound economics in a competitive vaccine market with three different formulations already approved for use. Indeed, a vaccine designed following these principles has just passed clinical trials (Platt et al, 2023).

**Use in future vaccination programmes** Multivalent pneumococcal vaccines are an established part of the childhood immunisation schedule and are being optimised post-hoc. Modelling not dissimilar to this is used to inform the content of seasonal flu vaccines (Hill et al, 2019). Although the pneumococcal model described above depended on data generated over the course of multiple vaccine introductions, future opportunities still exist for diseases where vaccines have not yet been introduced. A seven-valent vaccine against UTI-causing *Escherechia coli* is in clinical trials (Inoue et al, 2018), and a model of strain-dynamics based on the one described above tailored to this pathogen has been fitted already, and showed good results even without datasets spanning introduction of a vaccine (McNally et al, 2019). Also in clinical trials is a six-valent vaccine against group B streptococcus (Madhi et al, 2023), which has a similar strain and serotype structure to *S. pneumoniae*. Vaccines for group A streptococcus are still not available (Frost et al, 2023), but similar biology motivates the use of models and genomics for their design (Davies et al, 2019a).

For computational modelling to successfully enhance control of these pathogens the following things are needed:
- Availability of, and high-quality fitting to, historical genome and incidence data (Snape et al, 2012).
- Models which can accurately forecast strain dynamics, with this having been tested and verified in multiple datasets.
- Comparison of candidate models in terms of biological plausibility and prediction accuracy.
- Models which reflect the complexity of populations, using realistic numbers of strains and vaccine formulations, without becoming computationally intractable.
- Updating of model structures, data and advice over time, as the real effects of vaccination are observed on the pathogen population.

### Modelling within-host kinetics using individual-level longitudinal data

The immune system of an individual changes in response to an exposure event—whether pathogen infection, vaccination, or injection with monoclonal antibodies. Data from relevant clinical studies and trials can capture this change over time. Typical measurements include longitudinal observations of: neutralising antibodies, viral load, or specific immune markers (CD4$^+$ T-cell counts, IgG and IgM concentrations, viral load, etc.). However, data are typically sparsely distributed over time and contains measurement error, confounding and bias, making a single time course or simple averaging unsuitable to make estimates of kinetics. Combining such datasets with models of underlying post-exposure kinetics can account for these limitations, allowing accurate inferences from noisy and high-variance data. For example, modelling viral load over time is often performed so by fitting models to a correlate of pathogenic load: the cycle threshold (Ct) value, measured as an outcome from reverse transcription polymerase chain reaction (RT–PCR) tests. The lower the number of PCR cycles needed before the threshold required for a positive test, the higher the viral load in the sample being tested, providing a quantitative proxy of viral load. Viral load kinetic estimates can provide crucial epidemiological information on say a typical infectious period and therefore inform public health guidelines on isolation time.

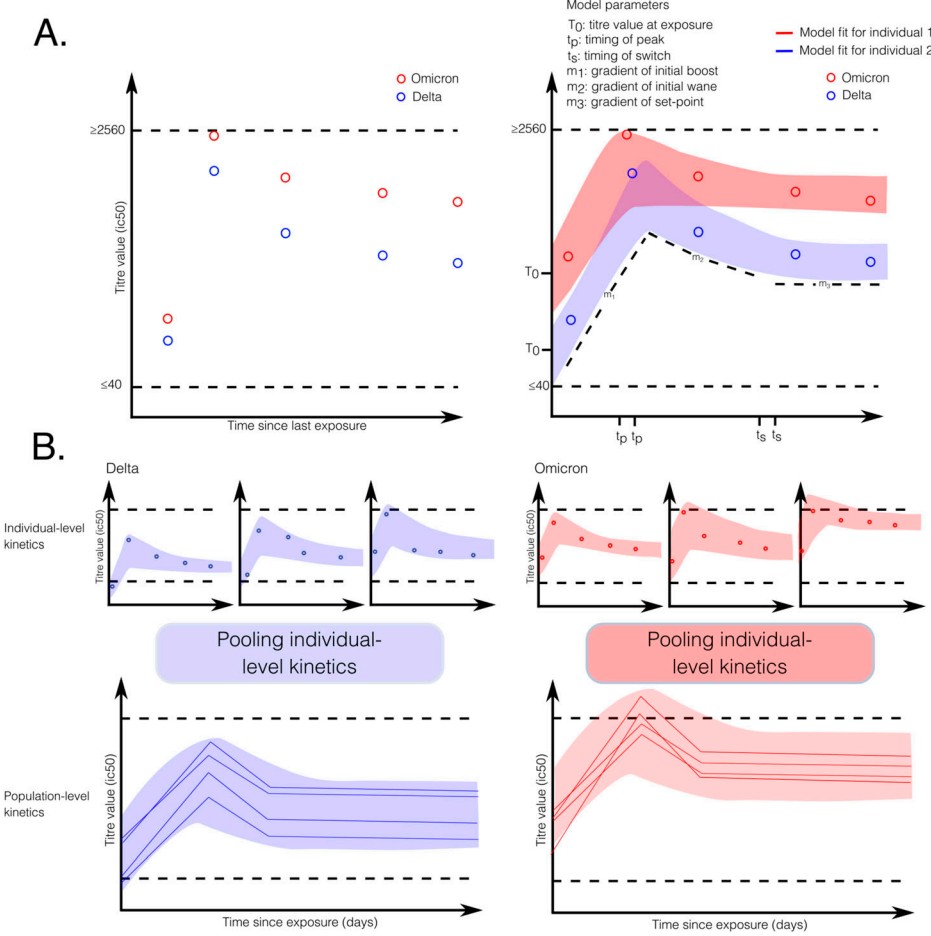

**Figure 3. Schematic of a typical antibody kinetics model, fit to longitudinal neutralising antibody data for SARS-CoV-2 stratified by two variants in a hierarchical framework.**
**(A)** A single individuals' neutralising antibody data (left) and resulting model fits (right), stratified by variant neutralised, using a typical six-parameter model of waning immunity.
**(B)** Hierarchical structure of the model. Top row: examples for three individual trajectories, split by variance. Bottom row: individual-level and population-level kinetics pooled for three individuals, giving more accurate population estimates.

However, a model is needed to integrate knowledge on measurement error, immune response, and individual variation. We present the general modelling framework, the technical and computational advances that were necessary to make such studies feasible, a number of recent detailed examples, and lastly some of the biggest outstanding challenges. Whereas our focus is on viral load and antibody kinetics, the approach can be applied to other biomarkers with knowledge of the immunological response.

**Hierarchical Bayesian model structure** Hierarchical Bayesian structures are typically used to model within-host kinetics because they generate more accurate inferences by pooling information between individuals. Furthermore, this model structure allows for the parameterisation of population-level priors using data from either individuals or populations. For instance, a study may be one study which report the incubation period for a pathogen as a probability distribution which could be used to parameterise a population-level prior in a hierarchical model. In the same model, estimates of individual-level variation in incubation period around this distribution can still be modelled without overly penalising individual outliers from the measured trends. This feature is useful in infectious disease modelling, where studies often focus on population-level or other higher level trends, but data come from

individuals. By parameterising priors at the appropriate level in the model, existing estimates inform the model estimates by allowing the information to flow through the hierarchy of parameters in a statistically robust manner. For example, consider a typical antibody kinetics model fit to longitudinal neutralising antibody data for SARS-CoV-2 stratified by two variants. The model structure includes individual-level trajectories (Fig 3A) and population-level kinetics pooled for multiple individuals (Fig 3B), providing more accurate population estimates.

**Technological and computational advances driving kinetics models** Late 20th century technological advances meant that the type of detailed individual-level datasets required for this type of modelling were more and more routinely collected. We outline the three most relevant advances to the examples presented:

The development of quantitative molecular techniques includes the RT–PCR test. RT–PCR allows for the rapid and highly sensitive and specific detection of viral RNA, which is essential for diagnosing infectious diseases like influenza and COVID-19. The speed and low cost at which RT–PCR tests can be performed and analysed means that large datasets of longitudinal RT–PCR test results are now often available, an advance significantly accelerated by and used during the COVID-19 pandemic.

Advances in laboratory technologies, including ELISA, flow cytometry, and mass spectrometry have improved the sensitivity and specificity of immunoassays used to detect antibodies and other immune markers. High-throughput sequencing technologies have revolutionised our ability to sequence DNA and RNA quickly and affordably, meaning that PCR test and immunoassay results can be stratified by the strain or even specific single-nucleotide variants of the relevant pathogen, creating more granular datasets (Wall et al, 2021a, 2021b; Pulliam et al, 2022; Wu et al, 2022; Cohen et al, 2024).

Using these advances has the potential to generate datasets that provide comprehensive views of variations in infectiousness itself or aspects of infectiousness (He et al, 2020) stratified by strains of infecting pathogen, type of vaccine administered, and other sets of covariates alongside longer term immune response data (Kucharski et al, 2018; Salje et al, 2018; Sun et al, 2022b; Wu et al, 2022; Russell et al, 2024).

Because of large datasets, potentially complex underlying dynamics, and high total numbers of parameters, these models are often computationally expensive to fit. Recent methodological advances have significantly improved computational efficiency for models with a high dimensional parameter space. The main advance relevant to the examples discussed here is the Markov Chain Monte Carlo sampling algorithm Hamiltonian Monte Carlo (HMC) (Betancourt, 2017). HMC uses the gradient of the parameter space to direct the trajectory to areas of high posterior density. As such, the parameter space is required to be differentiable and continuous, meaning discrete-valued parameters cannot be sampled from. Implementations of HMC samplers in a number of domain-specific languages such as stan means that the user no longer needs to write their own implementation of complex numerical methods or perform multidimensional calculus to fit such models.

**Recent examples of antibody kinetics models** A recent prominent example of modelling individual-level neutralising antibody kinetics reconstructed individual-level antibody kinetics after individuals have been exposed to one of four serotypes of Dengue fever (Salje et al, 2018). The study authors assume a standard antibody kinetics model structure including three phases: an initial rise in overall neutralising antibodies after infection, a waning phase, and a constant set-point dynamic corresponding to long-lasting immunity against the same serotype. They combine the kinetics model with a component able to estimate the probability that an infection event was missed. The individual-level kinetics are combined with the total numbers of infections to arrive at overall "protection over time" curves. These curves represent the population mean values of the protection against a specific serotype of Dengue fever, a crucial quantity for planning vaccination campaigns and public health interventions. Overall, if titres had been averaged and trajectories fit to the averaged data, the estimates would be far less accurate as to be useless—unmodelled noise between individuals would wash out all signals. Conversely, if individual-level trajectories had been fit without the hierarchical structure in place to share information, the model would say nothing about the level of immunity in the whole population.

New viral variants with either a higher measured peak of viral load or a longer duration of shedding correspond with higher viral transmission. Therefore, a number of key studies during the COVID-19 pandemic that measured changes in viral load as the pathogen evolved were key to assessing whether to expect an increase in transmission (Kissler et al, 2021, 2023; Challenger et al, 2022; Singanayagam et al, 2022). The specific question in each study varied following what was particularly crucial from a public health perspective at the time. However, the underlying model structures and datasets analysed were similar across all of these studies. All contained a viral kinetic model, fit to individual-level viral load or cycle threshold data; the statistical structure was hierarchical in each case, incorporating existing estimates of viral kinetics within the population-level priors; the kinetics model was either stratified by covariates included within the model structure, or the model was just fit separately to the dataset split up by individuals corresponding to the covariates of interest. Given the abundance of PCR and viral load data, relative to longitudinal neutralising antibody data, at this point, there are notably more viral kinetic studies at present.

However, traditional, pooled approaches also provided significant insights. For instance, Khoury et al (2021), demonstrated the utility of pooled data in identifying correlates of protection against COVID-19, whereas SARS-CoV-2 antibody landscape studies (Rössler et al, 2023; Wilks et al, 2023) highlighted the complexity of antibody interactions with different viral variants, offering valuable guidance for vaccine design. These examples underscore the complementary strengths of both individual-level and pooled data approaches in infectious disease modelling.

**Ongoing challenges in antibody kinetic modelling** An outstanding yet common challenge is how to deal with undetected infections. It is rare for a study tracking biomarkers longitudinally to detect every infection, especially given that many pathogens have asymptomatic infections, cause infections with varying levels of shedding, or cause some level of presymptomatic transmission (Fraser et al, 2004). Estimates of protection afforded by neutralising antibodies or some other immunological biomarker—often referred to as a correlate of protection—rely on accurate estimates of the number of total infections.

As such, estimating correlates of protection for a number of diseases presents an ongoing challenge (Krammer, 2021; Plotkin, 2023). Many studies have developed bespoke frameworks in which this is possible and attempts have been made to produce software tools to standardise approaches to this problem (Salje et al, 2018; Hay et al, 2020). However, no model yet exists which both incorporates missed infections and is also able to be fitted with a modern and efficient MCMC method, such as HMC. These more realistic models cannot be routinely and automatically fitted and require bespoke expert efforts for every new application.

# Discussion

What should a computational model of an infectious disease aim to do? Modelling can be used both to attempt to adequately represent complex biological systems or for the production of counterfactual scenarios designed to inform or even motivate policy. To put it

crudely, scientists are interested in revealing the biological mechanisms behind noisy observations whereas policy-makers care less about these details and more about accurate forecasting. The tension is unavoidable: using models to weigh up evidence for different hypotheses can, and often does, end up prioritising different models compared with searching for models with the best predictive performance.

As modelling has been incorporated into campaigns to control pathogen threats, it has seen the introduction of financial and logistical concerns beyond its traditional remit. Infectious disease modellers now model biology alongside economists, geographers, and logisticians. The long-term dynamics, now spanning years, of the system they model are often counted in the billions of dollars or millions of deaths. This is the kind of output required to prompt WHO or UN fact-finding inquiries into pathogens to gauge the level of threat they pose to humankind (Ranjbar & Alam, 2023; World Health Organization, 2023).

As this review has highlighted, the results presented in such reports are almost always the output of a process that passes output from models into further models. A model predicts the prevalence of AMR, a further model then uses this output to model the impact on human health, and then a further model then uses this output to model the economic cost. How do we propagate uncertainty through these chains of models? This is especially pressing when the output of the model is on a global scale, involving high mortality estimates that can be hard to contextualise. Propagating as much uncertainty as possible through chains of multiple models can lead to very wide prediction intervals in the final output. These intervals likely reflect the genuine uncertainty about what will happen in the future, but in our experience they are also unlikely to be popular with policy makers who have to use the output. The ideal solution might be to produce one large model going straight from the original data to the final output, but this might be both computationally challenging or difficult because of incompatible modelling approaches between disciplines.

Producing a single large and comprehensive model of a pathogen threat can lead to problems if there is genuine scientific disagreement on how certain aspects of a pathogen should be modelled. The structure of the single model should not rely entirely on the scientific predilections of whoever made it. We can link these issues clearly to the SARS-CoV-2 pandemic in the UK: the UK government advisory committee SAGE tackled this problem for medium-term pandemic trajectory predictions by asking for input from multiple modelling groups, reflecting a range of modelling approaches and scientific opinions. Should forecasting or scenario modelling hubs that synthesise modelling output be adopted for all types of pathogen threats? The output from these modelling hubs might involve weighting model predictions as a function of previous predictive performance. In our opinion, this would certainly involve a more rigorous attitude to model evaluation and forecast scoring than is usual for the field (Reich et al, 2022; Bosse et al, 2023).

The models that are the best at forecasting may not have a mechanistic structure that allows them to answer counterfactual questions (such as "by how much will mask mandates reduce transmission?"). This is an important type of question for policy-makers, so we must make mechanistic models for pathogens. How

good at forecasting mechanistic models should be is unclear. If we continue to consider SARS-CoV-2 (Knock et al, 2021; Barnard et al, 2022; Keeling et al, 2022), it is not immediately clear whether a mechanistic model being good at forecasting 2 wk into the future tells us anything about its accuracy when predicting the effect of counterfactual scenarios regarding, e. g., lockdowns. If the model produces a predicted pandemic trajectory for a counterfactual scenario where the imagined conditions of this scenario are very similar to what in fact occurred, then it seems reasonable to retrospectively treat the trajectory as a forecast and compare the output to what was observed in reality. Alternatively the same model could retrospectively produce a forecast by re-running using what did actually occur in terms of policy, population movement, vaccine coverage, and so on, as input and predicting the pandemic trajectory. Knowing how feasible these routes of model evaluation are is important for the next pandemic.

As the COVID-19 pandemic recedes, we must also learn from our collective experiences of computational modelling during this crisis: the question now is to figure out what we learned about modelling as a discipline during this emergency period, where the practice of modelling was temporarily transformed. Are forecasting hubs the best approach to collaboratively modelling AMR or vaccine impact? Have new standards of data quality and quantity been set by COVID-19 that can be replicated for other pathogens?

Considering the challenges highlighted both by COVID-19 and the models above, we think the next stages in research must take into account the following issues. Models must be freely shared, in common formats, so that they are reproducible and reusable by other researchers. More attention must be paid to modelling as a discipline in its own right, as opposed to treating modelling as a tool to be discretely applied to multiple scientific problems. This will help standardise the framework for comparing model quality, and bring together people working on similar modelling solutions to different applications. Finally, if models are to be used to shape public policy, especially during emergencies, then modellers must be permanently embedded in institutions making these recommendations. It is not sustainable to rely on academic modelling transforming itself every time there is an emergency situation.

## Supplementary Information

## Acknowledgements

JA Lees and J Hellewell were supported by the European Molecular Biology Laboratory.

### Author Contributions

JA Lees: conceptualization, supervision, and writing—original draft, review, and editing.
TW Russell: conceptualization, investigation, and writing—original draft, review, and editing.

LP Shaw: conceptualization, investigation, and writing—original draft, review, and editing.

J Hellewell: conceptualization, investigation, and writing—original draft, review, and editing.

## Conflict of Interest Statement

The authors declare that they have no conflict of interest.

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
