## [Reviewer comments · Life Science Alliance]

Life Science Alliance

Recent approaches in computational modelling for controlling pathogen threats

John A. Lees, Timothy Russell, Liam Shaw, and Joel Hellewell

DOI: <https://doi.org/10.26508/lsa.202402666>

Corresponding author(s): Joel Hellewell, European Bioinformatics Institute

Review Timeline:	Submission Date:	2024-02-19
	Editorial Decision:	2024-04-12
	Revision Received:	2024-06-05
	Editorial Decision:	2024-06-07
	Revision Received:	2024-06-11
	Accepted:	2024-06-13

Transaction Report:

April 12, 2024

Re: Life Science Alliance manuscript #LSA-2024-02666

Dr. Joel Hellewell
European Bioinformatics Institute
Wellcome Trust Genome Campus
Saffron Walden CB10 1SA
United Kingdom

Dear Dr. Hellewell,

Thank you for submitting your manuscript entitled "Recent advances and challenges in computational modelling for controlling pathogen threats" to Life Science Alliance. The manuscript was assessed by expert reviewers, whose comments are appended to this letter. We invite you to submit a revised manuscript addressing the Reviewer comments.

Thank you for this interesting contribution to Life Science Alliance. We are looking forward to receiving your revised manuscript.

Sincerely,

B. MANUSCRIPT ORGANIZATION AND FORMATTING:

Reviewer #1 (Comments to the Authors (Required)):

I enjoyed reading this paper, but it's a bit lacking a broad strategy/plan and perhaps a bit too long. As it's a review rather than a primary research paper, I'm ignoring the review format request above. Neither is this a line-by-line critique (though I would note that the paper does require careful proofreading in places). Rather, here are some fairly high-level thoughts as to concept and structure:

- At first sight, the paper presents itself as a post-COVID review of infectious disease modelling, with an initial focus on technical/computational challenges. But then it evolves into a slightly eclectic collection of three cases studies - within-host kinetic models, (multi-valent) vaccination modelling and AMR.
- The last two are perhaps more related to each other than the first - being both focused on bacterial infections, with reference to both multi-strain models and evolution.
- The paper is v long as well as being a bit diffuse. One option would be to restyle it as a review of challenges in multistrain modelling of bacterial pathogens and to put the review of within-host modelling into a different paper. On balance, I would recommend this - though the introduction and final discussion would need rewriting to have much more a focus on open challenges in the modelling of bacterial pathogens and their control (via either treatment or vaccination).
- Even if three sections are retained, I would still recommend the introduction and final conclusion be rewritten to be a bit more linked to the three case studies. I'd remove most of the references to COVID (and malaria) and focus on multi-strain pathogens from the start.
- If all three sections are to be retained, more could be done to unify across the three challenges and to highlight the conceptual relatedness - e.g. by focusing on the challenges of multiple strains/serotypes/variants.
- Khoury Nat Med should be cited in relation to COVID correlates of protection. I would also add refs to Derek Smith's COVID antibody landscape work, which nicely brings out the complexity of the interaction between different Abs and different viral variants.
- Again, if the first case study is retained, the "antibody kinetics" title of the first section might be broadened, to include statistical modelling of the relationship between (multivalent) Ab titres, viral diversity and protection. The viral and Ab kinetics bit is arguably less interesting (and less related to the other two case studies), though can still be mentioned. I should note that far more has been done modelling pathogen (mostly viral) kinetics than immune system (e.g. Ab) kinetics - largely because quantitation of virus is easier - and this should be mentioned.
- Lines 366-376 - a lot has been done on characterizing multistrain COVID Ab landscapes, including work examining optimal vaccine design - the work of Derek Smith and others might be cited
- Lines 442-454 - I found this hard to follow (why was that work a breakthrough?). It's also really quite detailed compared with most of the text.
- More generally, I'd recommend editing the vaccination section down a bit (e.g. by a page) so that it more matches the AMR section in terms of detail.
- "The existence of 'universal vaccines' would remove the need for this modelling" - I found this line a bit flippant (and would it really eliminate the need for multistrain models to capture bacterial population dynamics?). I would remove.
- Generally the AMR section was easy to read and engaging. But it seemed to jump from topic to topic (burden estimation, multistrain dynamics, drug cycling) without much justification. Similarly, the papers selected for examination in some cases seem a bit arbitrary (true also in the other sections). Maybe the section might start by highlighting the different areas relating to AMR where models have been used.
- Indeed, overall this paper would benefit from more signposting for the reader as to the strategy and conceptual framework the authors are bringing to the review.
- As mentioned above, the Discussion (and introduction) needs some work in my view - to be better linked to the three case studies and to bring out some broad conclusions common to all three - rather than jumping to talk about COVID again.

Reviewer #2 (Comments to the Authors (Required)):

In this manuscript, the authors review recent advances and challenges in infectious disease modelling, with a particular emphasis on data-driven approaches. They focus on three specific areas (within-host response, vaccine design and AMR), while

drawing parallels to advances made during the covid-19 pandemic response.

The article is interesting and informative and for the most part clear. I think the authors have done an excellent job of addressing a complex and extensive topic using well chosen case studies.

I have made some suggestions below that I think would help clarify parts of the manuscript. I do not think addressing all of these should be a necessary condition for publication, so the authors should feel free to ignore suggestions they do not find helpful.

The one point I do think it would be important to address is that, occasionally, the manuscript makes strong and/or general statements that are not backed-up with evidence or referenced. For example, the paragraph starting line 48 discusses the differences between historical and contemporary infectious disease modelling, and implies historical approaches were less focused on data. I am not sure this is true and think the authors do not provide sufficient support for this statement. Another example is the statement about policymaker's attitudes to uncertainty (line 746). The manuscript would benefit from the authors either providing more support for these types of statements or clarifying that they are of a more speculative nature.

Suggestions:

1. Line 78: the meaning of "within-sample" in this context was not obvious to me.
2. Line 106: I would interpret the wording here to mean that Ferretti et al used an IBM, which I don't think was the case?
3. Within-host kinetics section

I thought this section contained useful and interesting discussion and insights. However, I had to read this section multiple times to fully understand the points the authors were making. I think the section could benefit from some light re-structuring or re-phrasing. For example, here are some points I struggled with:

The section addresses both immune marker and viral load kinetics. I don't think this is a problem, but requires a bit more sign-posting. For example, I would have benefited from a few sentences at the beginning explaining why we are interested in these two measures and whether they tell us the same or different things about the disease (presumably infectiousness for viral load, and susceptibility for immune markers?) The text also sometimes switches between the two without highlighting the change. This was particularly noticeable with Fig 1, which is referenced in a paragraph about incubation period, which I assume relates to viral load, but the figure represents antibody titres.

On a few occasions, I had the impression key ideas were explained too late in the text. For example, PCR is introduced line 157, and then re-introduced with a very clear explanation line 201. Similarly, when HMC is first introduced (line 226), I felt like I was missing an intuition for why this is an advance - more details are given line 280. Similarly, line 236 gives a really nice explanation of antibody kinetics, which would have been helpful earlier in the text (for example when looking at Figure 1).

4. Figure 2: here, does the y-axis illustrate cases of disease or just prevalence in the population? This was a little confusing, because in all cases, we see vaccine-induced strain replacement to a similar extent. Is this meant to be lower in panel A (if the y-axis illustrates disease) or is the figure meant to illustrate the effect on prevalence rather than disease?
5. Line 444: I didn't understand the reference to the model in Corander et al. as a 5 parameter model. If the strength of FDS varies between genes, then the number of parameters would be at least equal to the number of genes? (I would count each equilibrium gene frequency as a parameter.)
6. Future vaccination programmes section: I think this section is very interesting and important. For me, a key issue here is that parametrising the pneumococcal FDS model is only possible because of the data available from multiple previous vaccine introductions. How feasible is a similar approach without this type of dataset?

Reviewer #1 (Comments to the Authors (Required)):

I enjoyed reading this paper, but it's a bit lacking a broad strategy/plan and perhaps a bit too long. As it's a review rather than a primary research paper, I'm ignoring the review format request above. Neither is this a line-by-line critique (though I would note that the paper does require careful proofreading in places). Rather, here are some fairly high-level thoughts as to concept and structure:

- *At first sight, the paper presents itself as a post-COVID review of infectious disease modelling, with an initial focus on technical/computational challenges. But then it evolves into a slightly eclectic collection of three cases studies - within-host kinetic models, (multi-valent) vaccination modelling and AMR.*
- *The last two are perhaps more related to each other than the first - being both focused on bacterial infections, with reference to both multi-strain models and evolution.*
- *The paper is v long as well as being a bit diffuse. One option would be to restyle it as a review of challenges in multistrain modelling of bacterial pathogens and to put the review of within-host modelling into a different paper. On balance, I would recommend this - though the introduction and final discussion would need rewriting to have much more a focus on open challenges in the modelling of bacterial pathogens and their control (via either treatment or vaccination).*
- *Even if three sections are retained, I would still recommend the introduction and final conclusion be rewritten to be a bit more linked to the three case studies. I'd remove most of the references to COVID (and malaria) and focus on multi-strain pathogens from the start.*
- *If all three sections are to be retained, more could be done to unify across the three challenges and to highlight the conceptual relatedness - e.g. by focusing on the challenges of multiple strains/serotypes/variants.*

We thank the reviewer for their comments on the overall framing and content of the review. As we re-read our first version in light of the input from both reviewers here, we do agree that there is too much back and forth between our chosen case studies and post-COVID modelling. In line with your 4th and 5th suggestions above, we have made changes to the structure of the introduction and discussion, to keep the flow clear and more clearly introduce the case studies. We have also rewritten the abstract to better reflect these changes.

While we definitely appreciate your suggestion regarding multistrain modelling as a change of topic – and indeed this was something we discussed when writing the review – we opted to keep these studies for the following reasons. We were asked to write about a very broad topic, and decided that rather than try and review such a large area, to instead focus on areas of our own expertise where we could bring more

insights. We also hope that by focusing on new and evolving topics across a range of applications, this will appeal to a wider readership. We also feel that some meaningful references to modelling during the COVID pandemic are necessary, as a review not considering this recent major event, where modelling was under the spotlight more than ever, would feel like it was missing key context. However, as noted above, we certainly agree the previous presentation did not bring out this thought process well enough, and the link between COVID and the topics presented in detail could seem tenuous. We believe the text we've added to the introduction brings together these topics more clearly in this version, and thank you again for helping us consider the broad direction more critically.

- Khoury Nat Med should be cited in relation to COVID correlates of protection. I would also add refs to Derek Smith's COVID antibody landscape work, which nicely brings out the complexity of the interaction between different Abs and different viral variants.

We agree with the reviewer and have added references to both Khoury and Smith, in the context of what we call the “pooled approach”. I.e., without individual-level modelled estimates. The added references appear in the a revised version of the second paragraph in the *Recent examples of antibody kinetics models* section:

New viral variants with either a higher measured peak of viral load or a longer duration of shedding correspond with higher viral transmission. Therefore a number of key studies during the COVID-19 pandemic that measured changes in viral load as the pathogen evolved were key to assessing whether to expect an increase in transmission (Kissler et al, 2021, 2023; Challenger et al, 2022; Singanayagam et al, 2022). The specific question in each study varied following what was particularly crucial from a public health perspective at the time. However, the underlying model structures and datasets analysed were similar across all of these studies. All contained a viral kinetic model, fit to individual-level viral load or cycle threshold data; the statistical structure was hierarchical in each case, incorporating existing estimates of viral kinetics within the population-level priors; the kinetics model was either stratified by covariates included within the model structure, or the model was just fit separately to the dataset split up by individuals corresponding to the covariates of interest. However, traditional pooled approaches also provided significant insights. For instance, Khoury et al. (2021) demonstrated the utility of pooled data in identifying correlates of protection against COVID-19, while SARS-CoV-2 antibody landscape studies (Wilks et al. 2023; Rössler et al. 2023) highlighted the complexity of antibody interactions with different viral variants, offering valuable guidance for vaccine design. These examples underscore the complementary strengths of both individual-level and pooled data approaches in infectious disease modelling.

- Again, if the first case study is retained, the "antibody kinetics" title of the first section might be broadened, to include statistical modelling of the relationship between (multivalent) Ab titres, viral diversity and protection.

We agree and have added more details on the Dengue example (Salje et al, 2018) describing how they generate modelled estimates against each of the four serotypes and are thereby statistically modelling the relationship between them. Furthermore, we discuss how they are able to go on to estimate *protection over time* curves within the population stratified by serotype.

- The viral and Ab kinetics bit is arguably less interesting (and less related to the other two case studies), though can still be mentioned. I should note that far more has been done modelling pathogen (mostly viral) kinetics than immune system (e.g. Ab) kinetics - largely because quantitation of virus is easier - and this should be mentioned.

We agree with the reviewer. As such, we have added the following sentence to the end of this subsection:

Given the abundance of PCR and viral load data, relative to longitudinal neutralising antibody data, at this point there are notably more viral kinetic studies at present.

- Lines 366-376 - a lot has been done on characterizing multistrain COVID Ab landscapes, including work examining optimal vaccine design - the work of Derek Smith and others might be cited

Similar to the above reply, we have added a reference to two of the most relevant Smith et al Ab landscape studies

...while SARS-CoV-2 antibody landscape studies (Wilks et al. 2023; Rössler et al. 2023) highlighted the complexity of antibody interactions with different viral variants, offering valuable guidance for vaccine design.

- Lines 442-454 - I found this hard to follow (why was that work a breakthrough?). It's also really quite detailed compared with most of the text.

- More generally, I'd recommend editing the vaccination section down a bit (e.g. by a page) so that it more matches the AMR section in terms of detail.

We appreciate this perspective, and agree that the description in this section was overly detailed. As suggested, we have edited this down by around a page, removing some of the specific methodological or biological details specific to this system, while trying to keep the general points coming across. In doing this, we have also addressed the comment about the details on lines 442-454.

- "The existence of 'universal vaccines' would remove the need for this modelling" - I found this line a bit flippant (and would it really eliminate the need for multistrain models to capture bacterial population dynamics?). I would remove.

Yes, we agree, and have removed this paragraph.

- Generally the AMR section was easy to read and engaging. But it seemed to jump from topic to topic (burden estimation, multistrain dynamics, drug cycling) without much justification. Similarly, the papers selected for examination in some cases seem a bit arbitrary (true also in the other sections). Maybe the section might start by highlighting the different areas relating to AMR where models have been used.

Thanks. We understand this criticism and have rewritten the section's opening. We now articulate the structure properly and have also moved some discussion that jumped the gun to the section summary. We've also made clear that this section doesn't aim at being comprehensive (since AMR modelling is a disparate cross-pathogen threat), but instead aims to highlight good work across three areas: calculating AMR incidence, explaining it, and informing action. We've added linking text throughout to better transition between sections.

- Indeed, overall this paper would benefit from more signposting for the reader as to the strategy and conceptual framework the authors are bringing to the review.

- As mentioned above, the Discussion (and introduction) needs some work in my view - to be better linked to the three case studies and to bring out some broad conclusions common to all three - rather than jumping to talk about COVID again.

See our first response above, where we address these overall points.

Reviewer #2 (Comments to the Authors (Required)):

In this manuscript, the authors review recent advances and challenges in infectious disease modelling, with a particular emphasis on data-driven approaches. They focus on three specific areas (within-host response, vaccine design and AMR), while drawing parallels to advances made during the covid-19 pandemic response.

The article is interesting and informative and for the most part clear. I think the authors have done an excellent job of addressing a complex and extensive topic using well chosen case studies.

I have made some suggestions below that I think would help clarify parts of the manuscript. I do not think addressing all of these should be a necessary condition for publication, so the authors should feel free to ignore suggestions they do not find helpful.

The one point I do think it would be important to address is that, occasionally, the manuscript makes strong and/or general statements that are not backed-up with evidence or referenced. For example, the paragraph starting line 48 discusses the differences between historical and contemporary infectious disease modelling, and implies historical approaches were less focused on data. I am not sure this is true and think the authors do not provide sufficient support for this statement. Another example is the statement about policymaker's attitudes to uncertainty (line 746). The manuscript would benefit from the authors either providing more support for these types of statements or clarifying that they are of a more speculative nature.

We thank the reviewer for their comments. Overall, we have made it more clear throughout the manuscript where what we are claiming is the opinion of the authors. We have also strengthened the section on the differences between historical and contemporary infectious disease modelling to provide references to specific examples that we think typify historical vs contemporary approaches

Suggestions:

1. Line 78: the meaning of "within-sample" in this context was not obvious to me.

We have made the meaning of this line more clear

2. Line 106: I would interpret the wording here to mean that Ferretti et al used an IBM, which I don't think was the case?

Thank you, we have clarified this sentence

3. Within-host kinetics section

I thought this section contained useful and interesting discussion and insights. However, I had to read this section multiple times to fully understand the points the authors were making. I think the section could benefit from some light re-structuring or re-phrasing. For example, here are some points I struggled with:

The section addresses both immune marker and viral load kinetics. I don't think this is a problem, but requires a bit more sign-posting. For example, I would have benefited from a few sentences at the beginning explaining why we are interested in these two measures and whether they tell us the same or different things about the disease (presumably infectiousness for viral load, and susceptibility for immune markers?) The text also sometimes switches between the two without highlighting the change. This was particularly noticeable with Fig 1, which is referenced in a paragraph about incubation period, which I assume relates to viral load, but the figure represents antibody titres.

We thank the reviewer for pointing out the structure of this section could have been clearer. To aid with this, we have made the scope of the entire section broader, by changing it to:

Modelling within-host kinetics using individual-level longitudinal data.

We then discuss the most common biomarkers that you might fit these types of models to:

Typical measurements include longitudinal observations of: neutralising antibodies, viral load, or specific immune markers (CD4+ T cell counts, IgG and IgM concentrations, viral load, etc).

Last, we have added a sentence to the end of the first paragraph of this section talking about our focus here, and the scope of the topic more broadly:

While our focus is on viral load and antibody kinetics, the approach can be applied to other biomarkers with knowledge of the immunological response.

On a few occasions, I had the impression key ideas were explained too late in the text. For example, PCR is introduced line 157, and then re-introduced with a very clear explanation line 201. Similarly, when HMC is first introduced (line 226), I felt like I was missing an intuition for why this is an advance - more details are given line 280. Similarly, line 236 gives a really nice explanation of antibody kinetics, which would have been helpful earlier in the text (for example when looking at Figure 1).

We thank the reviewer for pointing out that key concepts could have been introduced in a more structured manner. We now explain RT-PCR in the first paragraph of this section, we introduce antibody kinetics before Figure 3 (which used to be Figure 1) and we have moved the description of HMC to where the concept is first mentioned — in the *Technological and computational advances driving kinetics models* subsection, rather than later on.

4. Figure 2: here, does the y-axis illustrate cases of disease or just prevalence in the population? This was a little confusing, because in all cases, we see vaccine-induced strain replacement to a similar extent. Is this meant to be lower in panel A (if the y-axis illustrates disease) or is the figure meant to illustrate the effect on prevalence rather than disease?

Thank you for pointing this out – this was meant to represent prevalence rather than disease cases and we have updated the figure and caption to fix this issue.

5. Line 444: I didn't understand the reference to the model in Corander et al. as a 5 parameter model. If the strength of FDS varies between

genes, then the number of parameters would be at least equal to the number of genes? (I would count each equilibrium gene frequency as a parameter.)

We first note that we have rewritten this section to remove some of this detail as suggested by reviewer #1 above.

But in answer to your question: yes, this was unclear from our original description. The model uses three parameters to specify FDS strength across the accessory genes: strong selection force, weak selection force, and proportion under weak:strong. The equilibrium gene frequencies are taken from the data and are not fitted parameters. (This is a potential weakness of this model – the data needs to span a time and have enough samples to estimate these frequencies accurately). Perhaps we could have highlighted the difference between a fitted and fixed parameter in these models, but on balance felt (especially in light of the review comments received) this was probably too involved for this review.

6. Future vaccination programmes section: I think this section is very interesting and important. For me, a key issue here is that parametrising the pneumococcal FDS model is only possible because of the data available from multiple previous vaccine introductions. How feasible is a similar approach without this type of dataset?

We are glad this more forward-looking section was appreciated. This is an important point to clarify. One of the cited studies (McNally et al 2019) were able to fit the same model structure in a population and pathogen which had not been perturbed by vaccination, which we see as a promising sign. We have noted this explicitly in the text here. We also note in our summary points of requirements for this model that historical data from different populations are a requirement; also that models need to be updated as effects of vaccines start to affect the population.

June 7, 2024

RE: Life Science Alliance Manuscript #LSA-2024-02666R

Dr. Joel Hellewell
European Bioinformatics Institute
Wellcome Trust Genome Campus
Saffron Walden CB10 1SA
United Kingdom

Dear Dr. Hellewell,

Thank you for submitting your revised manuscript entitled "Recent advances and challenges in computational modelling for controlling pathogen threats". We would be happy to publish your paper in Life Science Alliance pending final revisions necessary to meet our formatting guidelines.

- please be sure that the authorship listing and order is correct
- please remove figures from the manuscript file and upload them as single files
- please add a Running Title and a Summary Blurb/Alternate Abstract to our system
- please add the Twitter handle of your host institute/organization as well as your own or/and one of the authors in our system
- titles in the system and manuscript file should match
- please add figure legends to the main manuscript text after the References section
- please add an Author Contributions section to your main manuscript text
- please add a Conflict of Interest statement to your main manuscript text
- please add callouts for Figures 2A-C and 3A-B to your main manuscript text

A. FINAL FILES:

B. MANUSCRIPT ORGANIZATION AND FORMATTING:

Sincerely,

June 13, 2024

RE: Life Science Alliance Manuscript #LSA-2024-02666RR

Dr. Joel Hellewell
European Bioinformatics Institute
Wellcome Trust Genome Campus
Saffron Walden CB10 1SA
United Kingdom

Dear Dr. Hellewell,

Thank you for submitting your Commissioned Review entitled "Recent advances and challenges in computational modelling for controlling pathogen threats". It is a pleasure to let you know that your manuscript is now accepted for publication in Life Science Alliance.

Again, congratulations on a very nice paper. I hope you found the review process to be constructive and are pleased with how the manuscript was handled editorially. We look forward to future exciting submissions from your lab.

Sincerely,
